# Identification of MAPK10 as a Candidate Gene for High Milk Production in Water Buffaloes Through a Genome-Wide Association Study

**DOI:** 10.3390/ani15172567

**Published:** 2025-08-31

**Authors:** Wangchang Li, Huan Chen, Duming Cao, Xiaogan Yang

**Affiliations:** 1Guangxi Key Laboratory of Animal Breeding, Disease Control and Prevention, College of Animal Science & Technology, Guangxi University, Nanning 530004, China; liwangchang@st.gxu.edu.cn (W.L.);; 2Hunan Provincial Key Laboratory of Animal Intestinal Function and Regulation, Hunan International Joint Laboratory of Animal Intestinal Ecology and Health, Laboratory of Animal Nutrition and Human Health, College of Life Sciences, Hunan Normal University, Changsha 410081, China

**Keywords:** buffalo, genome-wide association study, milk production traits

## Abstract

This study identifies key genetic markers and candidate genes, such as MAPK10, offering new insights into enhancing milk production in water buffaloes through targeted breeding strategies. These findings pave the way for more productive dairy buffalo populations, benefiting farmers and global dairy industries.

## 1. Introduction

China possesses abundant genetic resources of indigenous cattle breeds, characterized by high biodiversity and wide geographical distribution, making them a crucial component of global bovine (Bos spp.) genetic diversity [1]. The country currently maintains 26 water buffalo (Bubalus bubalis) breeds, including 24 native breeds and 2 introduced breeds. While water buffalo are occasionally used for meat production, their primary economic value lies in dairy output. Water buffalo milk, renowned for its rich trace element content, diverse fatty acid profiles, and enhanced digestibility, has garnered significant research interest and emerged as a focal point in China’s dairy industry [2].

Water buffalo milk exhibits superior economic value compared to conventional cow milk, owing to its unique nutritional profile (e.g., higher fat content, calcium density, and digestibility), positioning it as a premium product in the dairy market. However, the average lactation productivity of water buffalo (5–8 kg/day) remains substantially lower than that of Holstein dairy cows (25–30 kg/day) [3]. This disparity underscores the urgency of implementing scientific interventions, such as genomic selection and gene-editing technologies, to enhance milk yield in water buffalo. Addressing this challenge through targeted strategies could not only drive industrial-scale production but also reshape the supply dynamics of China’s dairy market [4,5].

The revolutionary application of molecular genetic technologies has transformed livestock breeding, with Single Nucleotide Polymorphisms (SNPs) emerging as the cornerstone of genomic research [6,7,8]. Whole-genome resequencing, powered by high-throughput platforms like Illumina, enables comprehensive profiling of genetic variation, offering precise molecular tools for livestock improvement. Building on this foundation, Genome-Wide Association Studies (GWAS) employ rigorous statistical frameworks—such as using three statistical models (cBLUP [9], GMATs [10], and BayesR [11]) and false discovery rate (FDR < 0.05) corrections—to significantly enhance the detection efficiency of candidate genes (e.g., MAK1P0) associated with lactation traits. This methodological advancement provides a critical basis for elucidating the molecular mechanisms underlying milk production in water buffalo and advancing targeted genetic breeding strategies.

## 2. Materials and Methods

### 2.1. Animal Welfare Statement

The protocol for this study was approved by the Attitude of the Animal Care and Welfare Committee of Guangxi University (GXU2019-021), Nanning, China.

### 2.2. Phenotypes and Animal Resources

This study collected 305-day milk yield (MY) data from 78 water buffaloes, including purebred Murrah buffaloes (imported from Pakistan) n = 21; purebred Nili-Ravi buffaloes (imported from India) n = 21; and Murrah × Nili-Ravi crossbred buffaloes (F1 or backcross generations) n = 36. All farmed buffalo samples mentioned above were obtained from the research farm of the Institute of Buffalo, Chinese Academy of Agricultural Sciences.

All individuals were multiparous cows (2–5 lactations) with complete lactation records standardized to a 305-day lactation period. The target trait was the 305-day milk yield (MY), used to evaluate inter-breed productivity differences and heterosis effects.

### 2.3. Sample Collection, Sequencing and Data Storage

Genomic DNA was extracted from water buffalo blood samples collected via caudal vein puncture using a vacuum blood collection system (Vacutainer, Shanghai, China). Blood was drawn from the caudal vein at the junction of the 4th and 5th caudal vertebrae (10 cm from the tail base) after disinfection with 75% ethanol. A 5 mL sterile syringe (24 G needle) was vertically inserted 0.5 cm into the skin, followed by blood collection using vacuum tubes. Genomic DNA extraction was performed using the phenol-chloroform method: anticoagulated blood was lysed with CTAB buffer (100 mM Tris-HCl, 1.4 M NaCl, 20 mM EDTA, 2% CTAB) supplemented with β-mercaptoethanol (0.2%) at 65 °C for 1 h. Proteins were removed by phenol/chloroform/isoamyl alcohol (25:24:1) extraction, followed by chloroform/isoamyl alcohol (24:1) purification. DNA was precipitated with isopropanol, resuspended in TE buffer, and quantified using NanoDrop (OD260/OD280 ratio 1.7–2.0). DNA integrity was assessed by 0.8% agarose gel electrophoresis (TAE buffer, 120 V, 30 min). Qualified DNA was fragmented and prepared for Illumina library construction, followed by paired-end 150 bp sequencing on the Illumina HiSeq 2000 platform at Genedenovo Biotechnology (Guangzhou, China) [10]. The sequencing data have been uploaded to (https://doi.org/10.6084/m9.figshare.28629623.v1, accessed on 1 April 2025).

### 2.4. Alignments and Variant Identification

Clean reads were aligned to the reference genome (UOA_WB_1) using BWA-MEM (v0.7.17) [12], followed by SNP detection with standardized pipelines including Samtools (v1.9), Picard (v3.1.1), and GATK (v4.0) [13,14]. The detected SNPs were subjected to hard filtering via GATK’s “VariantFiltration” module with the following criteria: QD (QualByDepth) < 2.0; FS (FisherStrand) > 60.0; MQRankSum (MappingQualityRankSumTest) < −12.5; ReadPosRankSum (ReadPosRankSumTest) < −8.0; MQ (RMSMappingQuality) < 40.0; SOR (StrandOddsRatio) > 3.0. This filtering strategy follows GATK’s recommended hard-filtering thresholds, ensuring the retention of statistically reliable SNPs for downstream analyses.

### 2.5. Variation Filtering

In genome-wide association studies (GWAS), rare alleles (minor allele frequency [MAF] < 0.05), high missingness rates (>30% genotype data missing), and multi-allelic variants may introduce biases. To mitigate these errors, we employed PLINK (v1.9) to filter detected SNPs according to the following criteria: exclusion of multi-allelic sites (retaining only bi-allelic SNPs); removal of SNPs with MAF < 0.05; and filtering SNPs with >30% missing genotype data (parameter: --geno 0.3) [15]. This quality control pipeline ensures data reliability for downstream association analyses. Following stringent filtering criteria, a refined set of 448,470 markers was retained, comprising 381,569 SNPs and 66,901 Indels.

### 2.6. Population Structure Analysis

GCTA software (v1.92.2) was employed to construct a genetic relationship matrix (GRM) based on genome-wide SNP data, followed by principal component analysis (PCA) to explore population structure and genetic relationships among samples [16]. Concurrently, SNP markers were pruned for linkage disequilibrium (LD) using Admixture software (v1.3): A PLINK-based pipeline applied a 50 kb sliding window step size and a 10 SNP window size, removing one marker from each pair with an LD coefficient (r^2^) > 0.2. This filtering retained 448,470 independent SNPs for subsequent population structure analysis. Finally, PopHelper software (v2.2.7) [17] was utilized to generate bar plots illustrating the ancestral composition proportions and genetic architecture of individuals within each subpopulation.

### 2.7. Genome-Wide Association Mapping

To effectively control for potential confounding factors (e.g., population structure and kinship) in the analysis, principal component analysis (PCA) and a genetic relationship matrix (GM) were incorporated as random effects [18,19]. Subsequently, genome-wide association studies (GWAS) were conducted using three models, cBLUP, GMATs, and BayesR, to identify associations between molecular markers and target traits in a mixed population. Candidate loci were identified based on a significance threshold adjusted for multiple testing (0.05/n, where n represents the total number of markers). Candidate genes were identified based on their proximity to candidate SNP loci within approximately 50 kb upstream and downstream regions (total ~100 kb genomic span). The NCBI-maintained dbSNP database [20] (Database of Single Nucleotide Polymorphisms) was utilized to determine whether these candidate loci corresponded to previously reported genetic variants (i.e., rsID-annotated sites in dbSNP) or represented novel SNPs. This approach integrates genomic positional association with database annotation to facilitate the functional characterization of candidate genes.

### 2.8. Candidate-Associated Gene Pathway Enrichment

The study performed systematic functional enrichment analysis on the identified candidate genes using the Kyoto Encyclopedia of Genes and Genomes (KEGG) and Gene Ontology (GO) databases integrated within the KOBAS (v3.0) platform [21], aiming to elucidate their potential involvement in biological pathways and molecular functions. Pathway enrichment analysis was performed to evaluate the enrichment of candidate genes in specific functions or pathways, with a significance threshold of *p*-value ≤ 0.05 used to determine statistically significant functional enrichment.

## 3. Results

### 3.1. Phenotypic Value Statistics of the Traits

Based on the histogram analysis of 305-day milk yield (MY) phenotypic distribution, the average milk production in water buffalo was 2321.3 kg (Figure 1), with approximately 15% of individuals classified as high yielders (MY > 3000 kg). The genetic and molecular characteristics of these high-yielding individuals constituted a core focus of this study.

### 3.2. Population Structure

To systematically evaluate genetic differentiation among water buffalo populations, unsupervised principal component analysis (PCA) was conducted to resolve population structure (Figure 2). Key findings include the following: Distinct Population Divergence (Figure 2A): Nili-Ravi (NB) and Murrah (MB) populations exhibited significant separation, indicating inherent genetic divergence between these taurine-derived breeds. Subpopulation Discrimination (Figure 2B,C): NB and MB populations achieved ~90% classification accuracy in PC1-PC2 space, whereas hybrid water buffalo (ZB) individuals showed marked clustering overlap with NB and MB, suggesting closer genetic relatedness between ZB and the two riverine populations. Principal Component Efficacy (Figure 2D): The first three principal components (PCs) captured 11.3% of total genetic variance, significantly outperforming subsequent components. This demonstrates that the top three PCs adequately represent lower-order genetic information.

To dissect the genetic structure and ancestral composition of water buffalo populations, Bayesian clustering analysis was performed across K = 2–9 subpopulations, with genetic ancestry proportions visualized via bar plots (Figure 3). Each color represents a genetic cluster (K value), with the y-axis showing individual ancestry proportions (0–1). At K = 3, Nili-Ravi (NB), Murrah (MB), and hybrid water buffalo (ZB) exhibit distinct population-specific signatures. Cross-validation errors (CV) identified K = 3 as optimal, with minimum CV error (0.801) (Figure 4). This aligns with PCA-identified structure (Figure 2), confirming three biologically meaningful genetic clusters.

### 3.3. Results of the Genome-Wide Associations

In our GWAS, we have identified a large number of SNPs associated with MY (Figure 5). We identified two statistically significant SNPs and four candidate genes within a 50 Kb range surrounding these loci that were associated with the trait MY (Table 1).

### 3.4. Kyoto Encyclopedia of Genes and Genome Pathway Analysis of Candidate Genes

To elucidate the biological roles of candidate genes, we performed Gene Ontology (GO) and Kyoto Encyclopedia of Genes and Genomes (KEGG) pathway enrichment analyses. GO results revealed 20 significantly enriched terms (FDR < 0.05) across three domains: biological process (BP, n = 9), cellular component (CC, n = 5), and molecular function (MF, n = 3). Notable BP terms included regulation of biological processes (GO:0048518), immune responses (GO:0002376), metabolism (GO:0008152), and cellular activities (GO:0009987) (Figure 6A). KEGG analysis highlighted 20 enriched pathways (Figure 6B, Appendix A), grouped into five functional categories: genetic/environmental information processing, cellular processes, organismal systems, and human diseases. Key pathways included the following: Environmental sensing: Hippo (ko04391), MAPK (ko04013), and ErbB (ko04012) signaling pathways. Cellular homeostasis: Apoptosis (ko04215) and mitophagy (ko04137). Immune regulation: Toll/IMD (ko04624), IL-17 (ko04657), Th1/Th2 differentiation (ko04658), and prolactin signaling (ko04917).

Our findings demonstrate that MAPK10, a central candidate gene, potentially modulates lactation performance through interactions with prolactin, IL-17, mitophagy, and MAPK signaling pathways. These insights uncover molecular mechanisms underlying dairy production and offer actionable targets for genetic improvement in buffalo breeding programs.

### 3.5. Significant Association of Milk Protein Content with SNP Validation

To validate the association between SNP NC_037551.1:16156790 (MAPK gene) and 305-day milk yield (MY) in water buffalo, genotyping-based histogram analysis was performed (Figure 7). Key findings include the T/T homozygous mutation exhibiting a significantly higher average MY (4008.8 ± 312.5 kg) compared to C/T heterozygotes (2712.3 ± 289.1 kg) and C/C wild type (2076.8 ± 265.4 kg) (One-way ANOVA, *p* = 0.0017).

## 4. Discussion

### 4.1. Molecular Genetic Structure

In genome-wide association studies (GWAS), population ancestry differences represent critical confounding factors that may lead to spurious associations by altering allele frequency patterns [22,23,24]. To address this challenge, we integrated principal component analysis (PCA), kinship matrix, and cross-validation error (CV error) methods to systematically resolve genetic structure. Although ZB shares partial genetic overlap with MB/NB, kinship, CV error, and PCA collectively support dividing the 78 buffalo cohort into three genetic clusters corresponding to Murrah (MB), hybrid water buffalo (ZB), and Nili-Ravi (NB) populations. These results establish a robust population structure framework for subsequent GWAS, effectively mitigating ancestry-related confounding effects.

### 4.2. Genome-Wide Association Analysis of Reproductive-Related Traits

In the association analysis of MY, we identified a total of four candidate genes (mitogen-activated protein kinase10, MAPK10; Zinc Finger Protein 84, ZNF84; Zinc Finger Protein 26, ZNF26; Zinc Finger Protein 605, ZNF605). The high milk yield in water buffaloes is a highly complex process that involves the coordinated action of multiple signaling pathways. Among the identified genes, some are associated with key pathways, including the prolactin signaling pathway, IL-17 signaling pathway, mitophagy pathway, and MAPK signaling pathway.

Previous studies have indicated that the prolactin signaling pathway and MAPK signaling pathway plays a crucial role in regulating cellular growth, development, cell cycle progression, and other functions, all of which contribute to MY efficiency [25,26,27,28]. We have identified the candidate gene MAPK10 as playing a significant biological role within the prolactin signaling pathway and MAPK signaling pathway. Specifically, MAPK10 (Mitogen-Activated Protein Kinase 10) is involved in regulating various intracellular signal cascades, including those related to prolactin secretion and cell proliferation. These processes are crucial for enhancing milk production in water buffaloes. For instance, MAPK10 can interact with key components of the MAPK signaling pathway, such as ERK and JNK transcription factors, to regulate cellular processes [29]. This interaction helps maintain cellular homeostasis and supports efficient lactation function. By modulating the activity of JNK, MAPK10 further influences the state of ERK proteins, controlling their nuclear localization and transcriptional activity [30]. This regulatory mechanism enables cells to respond appropriately to environmental signals, producing a variety of physiological effects, including promoting growth, differentiation, survival, and apoptosis.

### 4.3. The Role of MAPK10 in Enhancing Buffalo Milk Production: Current Findings and Future Directions

This study identified a regulatory locus within the MAPK10 gene through complementary GWAS methodologies, providing critical insights for enhancing water buffalo milk production. However, the findings primarily serve as a preliminary molecular genetic marker hypothesis for high-yield traits, largely constrained by the limited sample size (n = 78) which precluded precise and robust inferences. This study systematically interrogated multiple QTL databases (e.g., AnimalQTLdb and QTLbase) but found no previously reported QTLs associated with the identified locus (NC_037551.1:16156790). Although the analysis adopted stringent quality control measures—including permutation tests, Bonferroni corrections, and false discovery rate adjustments—to eliminate non-significant associations, the conclusions remain exploratory in nature [31,32]. Future investigations aim to systematically integrate these genetic determinants with functional genomics and multi-omics approaches to fully elucidate the mechanistic role of MAPK10 in buffalo lactation physiology.

## 5. Conclusions

This study conducted comprehensive whole-genome sequencing and genome-wide association analysis (GWAS) to identify genetic markers associated with high milk yield in water buffalo, ultimately detecting two significant SNP loci. Functional annotation revealed that NC_037551.1:16156790 (MAPK gene) could serve as a candidate molecular marker for high-yield traits. However, due to the limited sample size (n = 78), the precise functional mechanisms of this gene remain inadequately characterized. Future research will prioritize expanding the cohort size and incorporating functional genomics approaches—such as transcriptomic profiling—to validate the regulatory role of MAPK in buffalo lactation physiology and provide robust genetic evidence for breeding applications.

## Figures and Tables

**Figure 1 animals-15-02567-f001:**
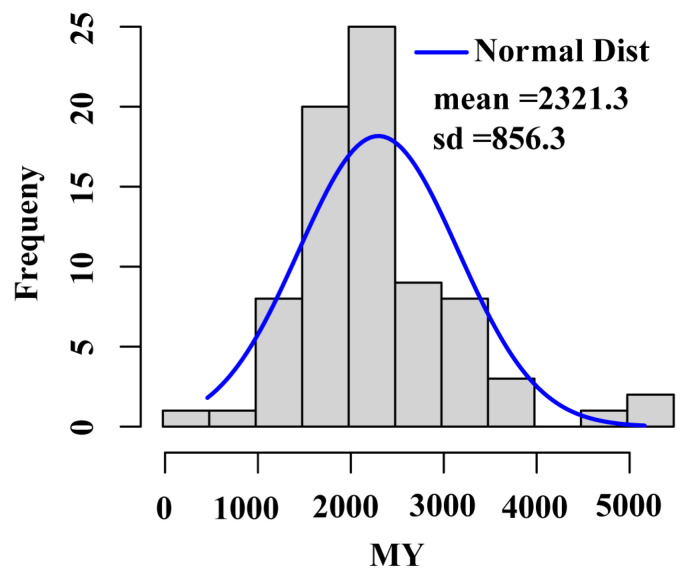
MY histogram. MY, 305-day milk yield.

**Figure 2 animals-15-02567-f002:**
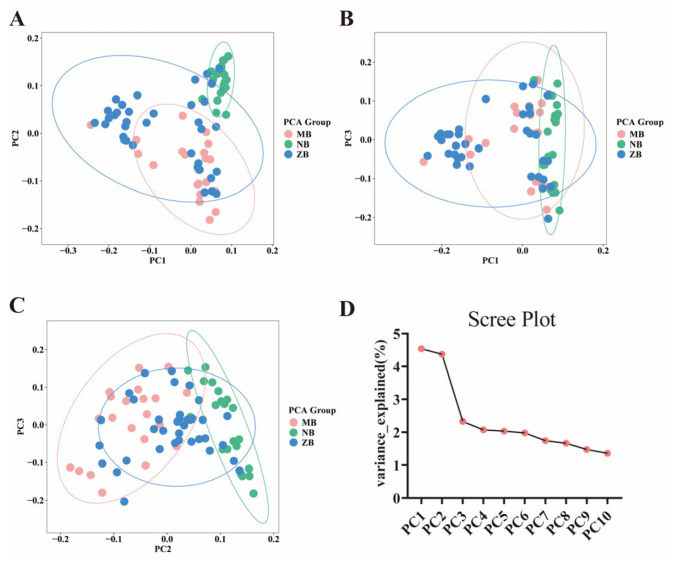
The population structure was visualized through three 2D PCA scatter plots (**A**–**C**) and one 3D PCA scatter plot (**D**), with the percentage of variance explained by each principal component (PC) indicated in parentheses (PC1: 28.5%, PC2: 19.3%, PC3: 14.7%). In the 2D plots, Panel A: PC1 vs. PC2 clustering; Panel B: PC1 vs. PC3 clustering; Panel C: PC2 vs. PC3 clustering. Each data point represents an individual buffalo, color-coded as follows: Murrah buffalo (MB): 21 individuals (pink); Nili-Ravi buffalo (NB): 21 individuals (green); hybrid water buffaloes (ZB): 36 individuals (green).

**Figure 3 animals-15-02567-f003:**
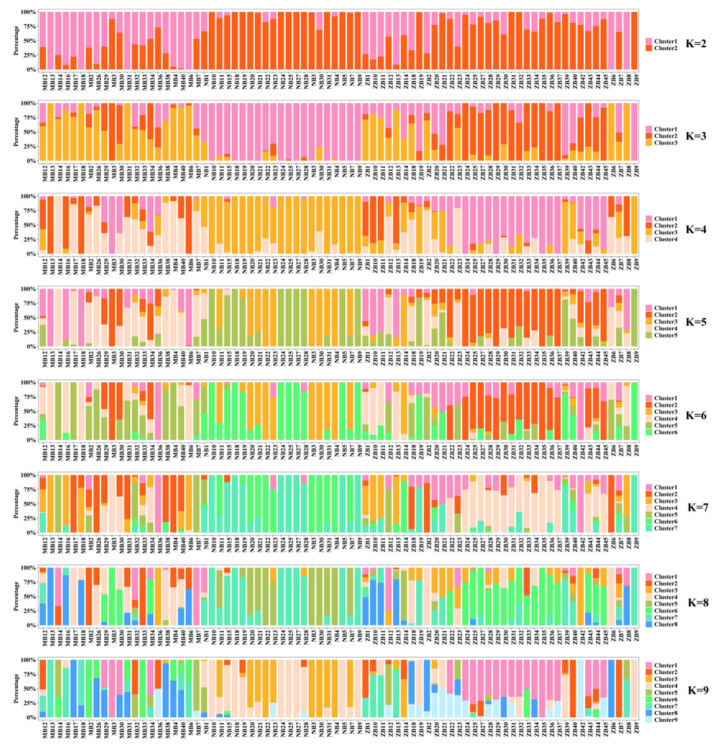
Genetic bar chart illustration for K-means clustering with varying numbers of clusters (K = 2 to 9).

**Figure 4 animals-15-02567-f004:**
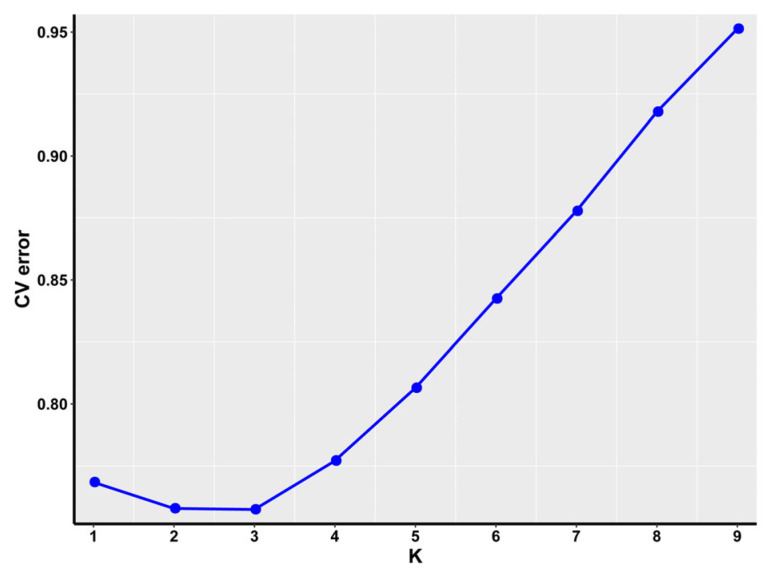
Line plot of cross-validation error rates across varying cluster numbers (K) in population structure analysis. The X-axis represents the number of clusters (K), and the Y-axis indicates the corresponding cross-validation error rate (CV error).

**Figure 5 animals-15-02567-f005:**
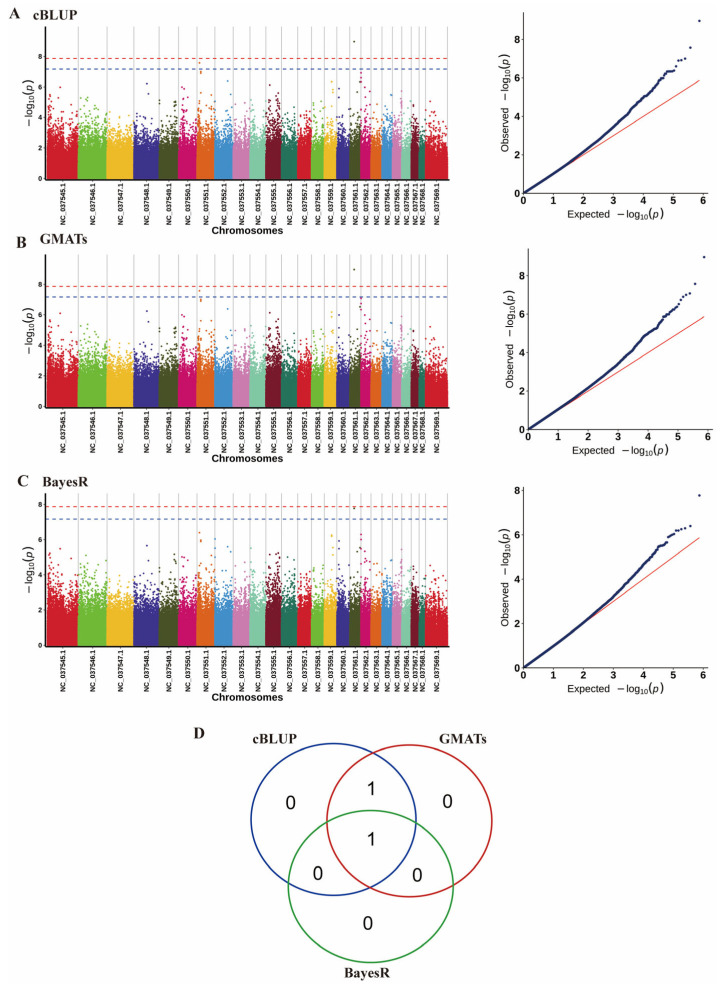
Association analysis of MY traits in water buffalo using three GWAS methods. (**A**–**C**) Comparative visualization of genome-wide association results from cBLUP, GMATs, and BayesR methods: Left panel (Manhattan plot) displays −log10(*p*) values for 30,145 SNPs across 25 chromosomes (24 autosomes + X). The blue horizontal line indicates a significance threshold of 0.05/n, The red horizontal line indicates a significance threshold of 0.01/n. SNPs exceeding this threshold are considered significant. Right panel (Q-Q plot) compares observed (−log10 *p*) values (x-axis) against expected uniform distribution values (y-axis). (**D**) Venn diagram summarizing overlapping significant SNPs identified by all three methods.

**Figure 6 animals-15-02567-f006:**
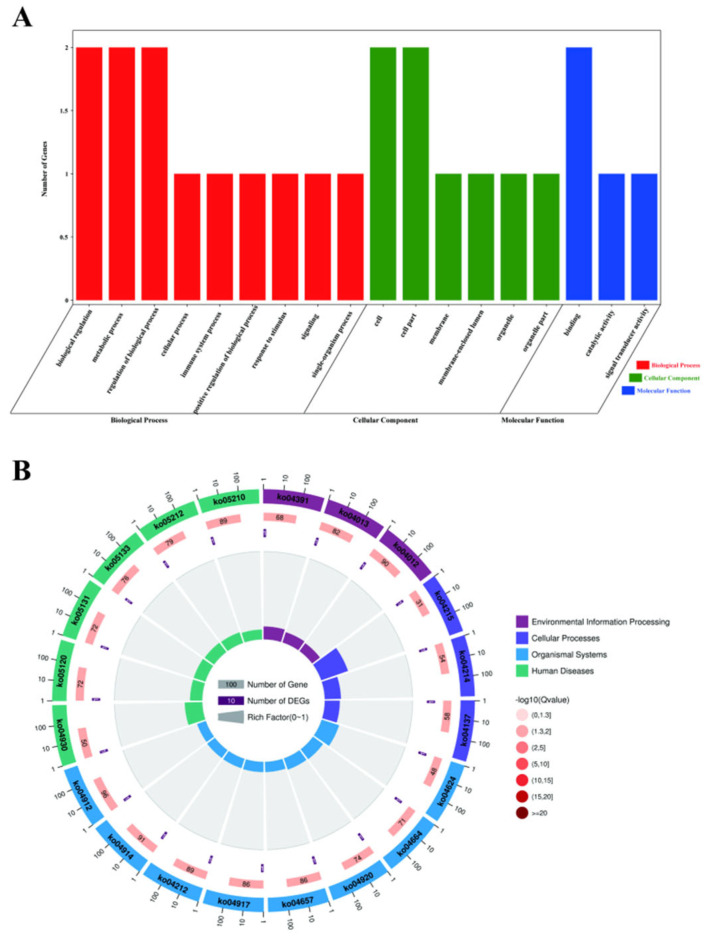
GO and KEGG enrichment analysis of candidate genes. (**A**) Top 20 significantly enriched Gene Ontology (GO) terms across three domains, cellular component, biological process, and molecular function, visualized as a horizontal bar chart. (**B**) Four-layer KEGG pathway enrichment plot: Outermost layer: Functional classification of enriched pathways, with scale bars indicating gene counts. Second layer: Gene counts and adjusted *p*-values (FDR) for each background category. Longer bars represent higher gene counts; red intensity correlates with statistical significance (smaller *p*-values). Third layer: Total candidate gene counts per pathway. Innermost layer: Enrichment factor (candidate gene count in category/background gene count per category), with grid lines spaced at 0.1 intervals. Color gradients reflect enrichment significance (darker shades = higher significance).

**Figure 7 animals-15-02567-f007:**
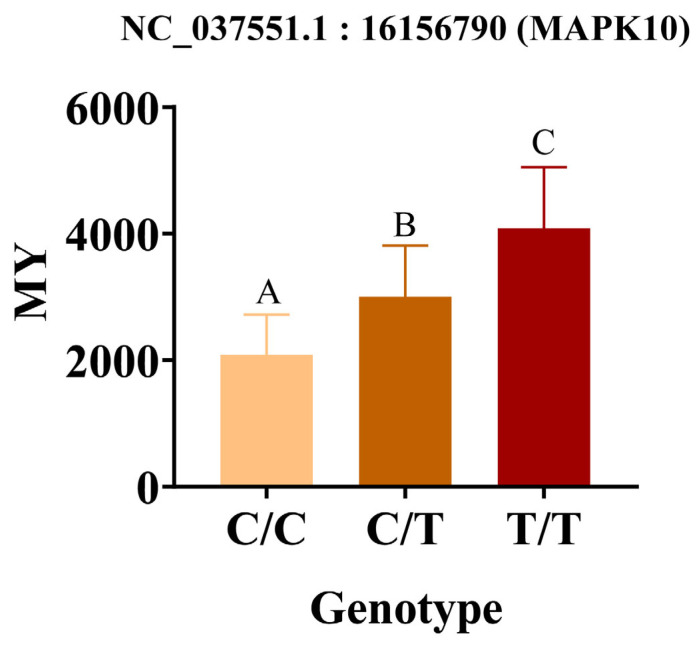
The phenotypic values of milk protein content. Different letters in the same column indicate significant differences (*p* < 0.05); T/T (homozygous mutation) type; C/T (heterozygous mutation); C/C (reference genotype) type.

**Table 1 animals-15-02567-t001:** The SNPs identified via genome-wide association analysis encompass detailed information *.

Methods	SNP	Chr	Pos	*p*	Candidate Genes
cBLUP	1	NC_037551.1	16,156,790	3.8 × 10^−8^	*MAPK10*
cBLUP	2	NC_037561.1	28,948,029	1.8 × 10^−9^	*ZNF84*, *ZNF26*, *ZNF605*
BayesR	1	NC_037551.1	28,948,029	2.8 × 10^−8^	*ZNF84*, *ZNF26*, *ZNF605*
GMATs	1	NC_037561.1	16,156,790	1.4 × 10^−9^	*MAPK10*
GMATs	2	NC_037561.1	28,948,029	1.5 × 10^−8^	*ZNF84*, *ZNF26*, *ZNF605*

* Mitogen-activated protein kinase10, MAPK10; zinc finger protein 84, ZNF84; zinc finger protein 26, ZNF26; zinc finger protein 605, ZNF605.

## Data Availability

https://doi.org/10.6084/m9.figshare.28629623.v1, accessed on 1 February 2025.

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
