# Peer review of "Identification of MAPK10 as a Candidate Gene for High Milk Production in Water Buffaloes Through a Genome-Wide Association Study"

_animals, 2025, doi:10.3390/ani15172567_

Round 1
Reviewer 1 Report
Comments and Suggestions for Authors
This paper provides a thorough analysis on the Role of MAPK10 in prolactin signaling and cell proliferation linked to high milk production in water buffaloes. These findings pave the way for more productive dairy buffalo populations, benefiting farmers and global dairy industries. This work paves the way for more targeted and effective breeding programs in the future. However, upon reviewing, I suggest minor revisions to further enhance the overall quality of the paper.
- Keywords: Buffalo; Milk production traits; Genome-wide association study (Arrange alphabetically)
- In the Introduction, since this paper focuses on buffalo, it would be beneficial to include relevant census data on the buffalo population in China such as the number of recognized breeds, total population, and their distribution. This context would strengthen the background and highlight the significance of the study.
- Several previous studies have been conducted on this topic; however, the authors have not cited or discussed any of them in the Introduction. Including a brief review of relevant literature would help position the current study
- In material and method section, please bold the subheadings also as it is difficult to highlight those sections.
- Samples were collected from field or any institutional farm, kindly clarify the origin of buffalo taken as sample.
- Kindly add the recent versions of utilized software to facilitate researchers and readers.
- Authors can also add some more analyses such gene-gene interaction, etc.
- Displays the sample clustering obtained from PCA through three two-dimensional scatter plots, but authors have not mentioned the significance of principal components 1 and 2. Kindly elaborate.
- Authors have not mentioned associated QTLs from the selected genes. They have not fine it or not done it? Please clarify
- The authors have not mentioned any associated QTLs for the selected genes. It is unclear whether this analysis was not performed or was performed but not reported. Clarification on this point is needed
Lack of a conclusion: The paper concludes without summarizing the main findings, recommendations for further research, or practical lessons learned. Provide a significant conclusion that summarizes the key conclusions and recommendations.
Author Response
We have already made revisions based on your valuable suggestions and provided scientific one-on-one responses. The relevant reply content has been uploaded in the attachment. Please check it in time.

Reviewer 2 Report
Comments and Suggestions for Authors
This study, through a systematic design, identified a potential association between the MAPK10 gene and high milk yield traits in buffaloes. The authors employed a genome-wide association study (GWAS) combined with dual genotyping strategies, preliminarily suggesting statistical linkage (P < 0.05) between MAPK10 loci and milk production traits. This work provides exploratory insights into the genetic basis of buffalo lactation traits and offers potential targets for genomic selection in breeding programs, though practical application necessitates validation in large-scale populations. Despite the high quality of the work, several aspects require clarification or could be strengthened to make the conclusions more robust.
1. Sample Size Justification. The most apparent limitation is the small sample size. Although the selection of extreme phenotypes (upper and lower quartiles) enhances the biological signal—likely contributing to statistically significant results—such a small group is sensitive to individual variation. The authors should explicitly acknowledge this limitation in the "Discussion" section. It is recommended that they emphasize that, despite the statistical significance, the findings should be considered preliminary and require validation in a larger cohort. This will enhance the objectivity and credibility of the study.
2. Investigation of Other Genes. The study identified 4 candidate genes. While the focus was placed on genes within the MAPK signaling pathway, what about the rest? What other biological processes might be fine-tuned through candidate genes? An expanded analysis of the candidate gene list may uncover parallel or complementary regulatory pathways involved in proliferation function.
Author Response

(The authors gave the same response as above.)

Reviewer 3 Report
Comments and Suggestions for Authors
The manuscript presents several valuable findings regarding population structure and the genetics of milk yield traits in water buffaloes.
However, there are several areas that require attention:
-
Text Overlap: A significant portion of the text—approximately 50%—overlaps with previously published work. This level of similarity is unacceptable and must be addressed.
-
Title Relevance: The title does not accurately reflect the key findings or the overall content of the manuscript. While the authors suggest a potential link to the MAK10 gene, the evidence is insufficient to confirm its role, particularly through the prolactin signaling pathway in milk production. Further functional studies are needed to substantiate this claim.
- Sample Size and Reliability: The sample size used in the study is quite limited, which raises concerns about the reliability and robustness of the results at this stage.
- The methods were used as pretty the same as the previous paper for GWAS. It is not necessary to have all three GWAS methods based on similar assumptions, if the authors would like to compare the methods as well, please select some Bayesian based methods, or machine learning based approaches. "GRM1 as a Candidate Gene for Buffalo Fertility: Insights from Genome-Wide Association Studies and Its Role in the FOXO Signaling Pathway".
The authors need to extensively edit the manuscript.
Author Response

(The authors gave the same response as above.)
